# A Proteomic Examination of Plasma Extracellular Vesicles Across Colorectal Cancer Stages Uncovers Biological Insights That Potentially Improve Prognosis

**DOI:** 10.3390/cancers16244259

**Published:** 2024-12-21

**Authors:** Abidali Mohamedali, Benjamin Heng, Ardeshir Amirkhani, Shivani Krishnamurthy, David Cantor, Peter Jun Myung Lee, Joo-Shik Shin, Michael Solomon, Gilles J. Guillemin, Mark S. Baker, Seong Beom Ahn

**Affiliations:** 1Macquarie Medical School, Faculty of Medicine, Health and Human Sciences, Macquarie University, Sydney, NSW 2109, Australia; abidali.mohamedali@mq.edu.au (A.M.); benjamin.heng@mq.edu.au (B.H.); shivani.krishnamurthy@hdr.mq.edu.au (S.K.); mark.baker@mq.edu.au (M.S.B.); 2School of Natural Sciences, Faculty of Science and Engineering, Macquarie University, Sydney, NSW 2109, Australia; 3Australian Proteome Analysis Facility, Macquarie University, Sydney, NSW 2109, Australia; ardeshir.amirkhani@mq.edu.au (A.A.); david.cantor@mq.edu.au (D.C.); 4Department of Colorectal Surgery RPAH & Institute of Academic Surgery, Sydney Medical School, University of Sydney, Sydney, NSW 2050, Australia; peter.lee1@health.nsw.gov.au (P.J.M.L.); professor.solomon@sydney.edu.au (M.S.); 5Department of Tissue Pathology and Diagnostic Oncology, Royal Prince Alfred Hospital, Camperdown, Sydney, NSW 2050, Australia; jooshik.shin@health.nsw.gov.au; 6Department of Chemistry, Faculty of Mathematics and Natural Sciences, Institut Pertanian Bogor University, Bogor 16680, Indonesia; gilles.guillemin@apps.ipb.ac.id

**Keywords:** colorectal cancer, extracellular vesicles, plasma microvesicle proteins, exosomes, protein biomarkers, SWATH-MS

## Abstract

This study investigates the role of plasma extracellular vesicles (EVs) in understanding colorectal cancer (CRC) progression and recurrence to potentially improve disease prognosis. Using advanced SWATH-MS proteomics, EV proteins from CRC patients across four stages and healthy controls were analysed. This study identified 11 key proteins differentiating early-stage CRC from healthy individuals and 14 proteins associated with tumour recurrence risk. These proteins reflect changes in cancer-related processes, including metabolism, cytoskeletal remodelling, and immune response. The findings provide valuable insights into the role of EVs in CRC and emphasise the need to standardise methods for EV isolation and analysis to enhance biomarker discovery and allow seamless clinical applications

## 1. Introduction

Colorectal cancer (CRC) is the third most prevalent cancer globally, with high mortality rates due to the lack of accurate and easily accessible early-stage diagnostic/prognostic tools [1]. However, if detected early, CRC is often curable following surgical resection, leading to an excellent 5-year survival rate (>90%) [2]. Nevertheless, current trends show that approximately 5–33% of patients who have had surgical resection ultimately experience tumour relapse within 5 years—5% for stage I, 12% for stage II, and 33% for stage III [2,3]. Of these, ~10–20% occur at a distance from the original tumour and are often not visible within the lumen of the gastrointestinal tract, suggesting that some form of micro-metastasis had already occurred [4] prior to surgery. Therefore, it is unsurprising that significant efforts continue to be directed towards effective screening strategies to identify early-stage CRC (diagnostic) and, most importantly, prognostic indicators to predict tumour relapse with a view to further decrease mortality [5].

The faecal occult blood test (FOBT) and the faecal immunochemical test (FIT) both detect microscopic amounts of occult blood in faeces/stool. These are the most commonly used CRC population screening tests globally [6]. More recently, multitargeted stool DNA [MT-sDNA] or FIT-DNA tests have demonstrated better sensitivity for advanced malignant lesions but not for pre-cancer advanced adenomas [7]. Despite the high sensitivity of the test for occult blood, compliance rates for stool tests are relatively low (40–56%) due to participants *‘not wanting to handle their own stool’* or *‘not wanting to keep stool samples on a card in the house’*. These have been rated among the top five barriers to these types of screening programmes [6,8]. Additionally, faecal tests have high false positive rates (low accuracy) for detecting CRC, as gastrointestinal bleeding may be associated with other common non-malignant conditions (e.g., colitis, anal fissures, and haemorrhoids) [9]. Developing a sensitive and accurate test with a high patient compliance rate (e.g., blood-based biomarker test) would solve a significant and unmet clinical need.

Blood plasma and/or serum are routinely collected for diagnostic analysis of proteins/peptides and other biomolecules (RNA, ctDNA, etc.). A subset of these plasma biomarkers originate from extracellular vesicles (EVs) [10,11] and have been considered entities for diagnosis and prognosis and indicators of therapeutic response in human disease [12]. These vesicles contain biologically active material that is representative of the parental cell of origin and are thought to participate in cell-to-cell communication, resulting in changes to the biology of the recipient (including distant) cells [11,12,13]. Blood plasma EVs are commonly classified into three principal populations based on their intracellular origin, namely, (i) apoptotic bodies, (ii) exosomes, and (iii) microparticles (MPs) (also known as microvesicles (MVs, ectosomes or shed microvesicles)) [14,15]. The formation of apoptotic bodies (size: 50–5000 nm) is a consequence of cell death by apoptosis [16,17]. Conversely, exosomes (size: 30–100 nm) and MPs (size: 50–1000 nm) are released at all stages of cell growth even though the mechanisms and machinery involved in their formation differ [18]. Exosomes are generated by the formation of intraluminal vesicles (ILVs) in endosomal compartments or multi-vesicular bodies. Multi-vesicular bodies containing ILVs translocate to the cell periphery and fuse with the plasma membrane to release their ILVs, which are then considered exosomes [19].

There is growing evidence that EVs may be instrumental in various processes in cancer development [20], including invasion, migration, metastasis, and cancer immune modulation. It is thought that EVs, by transporting surface-anchored proteins and cargo, affect different cells in the tumour microenvironment [21,22]. Additionally, actively metastasising cells have been shown to shed a larger quantity of EVs into surrounding tissues [23]. With the development of EV protein profiling using mass spectrometry (MS), EV proteomics allows the identification of proteins that are over/under-expressed in tumour-derived EVs, and these can be used in cancer diagnosis/prognosis, building a better understanding of cancer biology—both guiding appropriate clinical intervention and contributing to better patient management [24]. Proteomic research in CRC EVs is accelerating with several studies demonstrating the efficacy of using proteomics technologies to elucidate protein biomarkers of CRC (Table 1), albeit using different approaches and, interestingly, with different results.

Recognition of the exploration of EVs has been demonstrated by the development of two large manually curated EV databases—Exocarta [25] and Vesiclepedia [26]—since 2012. These databases act as repositories of EV cargo and membrane protein data from various experiments performed at various qualities and stringencies using a variety of technologies and methodologies. However, as more data is added, they serve to increase confidence in the most frequently observed EV markers. Furthermore, as part of the Human Proteome Project (HPP), the human plasma PeptideAtlas 2021-06 database released a human extracellular vesicle PeptideAtlas 2021-06, which consisted of 2757 canonical proteins detected in EVs circulating in the blood [27] and detected using a universally accepted high-stringency analysis pipeline.

Here, we utilised a highly curated homogeneous cohort of CRC patient plasmas (n = 80, with 20 from each of the four CRC stages I–IV) and healthy controls (n = 20) from whom EVs were isolated/derived with the aim of discovering markers for both early detection of CRC as well as the risk of tumour relapse within five years from CRC resection surgery. We hypothesised that a simple and robust EV isolation method would be sufficient to enable comprehensive proteomic analysis and facilitate the discovery of reliable biomarkers for CRC detection and prognosis. These plasma samples are identical to those used in our previous plasma protein biomarker studies [28,29,30]. To achieve our aims, we isolated EVs using an effective centrifugation technique and subsequently employed SWATH-MS (Sequential Window Acquisition of all THeoretical Mass Spectra), as it has a unique ability to achieve deep proteomic coverage, reproducibility, and reliable quantification across samples [31,32]. Unlike traditional DDA (data-dependent acquisition) approaches, SWATH-MS captures all detectable peptides within predefined mass ranges in a data-independent manner, reducing bias toward highly abundant proteins [33]. This method is particularly advantageous for complex biological samples, such as plasma-derived EVs, where a comprehensive and reproducible protein profile is critical for biomarker discovery [34].

We found 853 total proteins in isolated EVs from normal and CRC plasma using HPP high-stringency protein inference parameters (i.e., each protein identified with ≥2 non-nested neXtProt proteotypic peptides of ≥9 amino acids each). The identified proteins demonstrated the efficacy of our experimental approach after comparison against EV databases (Exocarta, Vesiclepedia, and the more recent PeptideAtlas study). Furthermore, we discovered 11 differentially regulated proteins (i.e., ALDH9A1, F9, SMIM1, ADIPOQ, C1QC, ACTN2, HPR, LRRFIP1, CYRIB, PFKP, and MDH2) between stage I CRC compared to healthy controls. We also found 12 differentially regulated proteins (i.e., TMED9, NSF, COL6A1, ENO2, DARS1, LRRFIP2, PGM1, GDI1, SLC2A1, IQGAP1, CAMK1, and TSG101) in the cumulative set of CRC stages I, II, and III patients who exhibited tumour recurrence compared to patients with no evidence of recurrence. We compared our results to previously published studies (Table 1), considered the limitations of our study, and discussed critical issues regarding the observed lack of concordance in other proteomics EV studies on CRC.

**Table 1 cancers-16-04259-t001:** Review of blood extracellular vesicle protein biomarkers for colorectal cancer using MS-based proteomic approaches on human biofluids.

Author (Year) [Ref]	Sample	Sample Size	EV Isolation Method	Analytical Technique	Biomarker (Gene Name)	Clinical Utility	Criteria
Choi et al. (2011) [35]	CRC ascites	3 CRC ascites	Sucrose gradient and OptiPrep density gradient centrifugation	SDS-PAGE, LC-MS/MS	CD97, CD9, TSPAN8	CRC diagnosis	1% FDR protein level
Chen et al. (2017) [36]	CRC serum	18 CRC *, 31 HC * Validation on 18 CRC and 18 healthy controls.	Ultracentrifugation	LC-MS/MS, TMT label	AHSG, FN1, HSP90AA1	CRC diagnosis	5% FDR peptide level, one peptide 7 ± amino acids
Shiromizu et al. (2017) [37]	CRC serum	16 CRC (stage II and IV) and 8 HC (Shotgun). SRM validation 56 CRC, 28 HC.	Sucrose gradient centrifugation	Literature, Shotgun LC-MS, Validation by SRM	ANXA3, ANXA4, ANXA11	Pre- and post-metastasis CRC	1% FDR protein and peptide level
Menck et al. (2017) [38]	CRC Plasma	Overall, 330 cancer patients, of whom 28 were CRC	Ultracentrifugation	Western blotting	EMMPRIN	CRC diagnosis	N/A
Lee et al. (2018) [39]	CC plasma	46 CC and 33 HC. Discovery study on HCT29 cell lines.	Differential centrifugation	LC-MS/MS	TSPNA1	CRC diagnosis	2 ± peptides with FDR below 2%
Zhong et al. (2019) [40]	CC serum	78 stage III CC *, 40 HC *	Ultracentrifugation or Total Exosome Isolation Reagent	LC-MS/MS, TMT label	SPARC, LRG1	CC diagnosis, recurrence prediction	2 unique peptides (7 ± amino acids)
Zheng et al. (2020) [41]	CRC plasma	Discovery *: 10 CRC with/without LM (n = 20), 10 HC Validation **: 13 adenomas, 12 CRC without LM, 12 HC	Sucrose gradient centrifugation	LC-MS/MS, DIA, PRM	FN1, S100A9, HP, FGA	CRC diagnosis	
Chang et al. (2021) [42]	CRC serum	12 non-AAs, 13 AAs, 16 stage I CRC, 15 stage II CRC, 16 stage III CRC, 15 stage IV CRC, 13 HC	Size exclusion chromatography	LC-MS/MS	GCLM, KEL, APOF, CFB, PDE5A, ATIC	CRC diagnosis	1% FDR, 1 razor + unique peptides (6 ± amino acids)
Lin et al. (2022) [43]	CRC Serum	Discovery: 56 CRC patients with CRLM compared to 7 with benign liver disease (BD). Validation: 154 CRLM, 78 BD (internal) and 110 CRLM (external)	N/A	N/A	CD14, LBP, CFP, Serpin A4, CXCL7	CRC prognosis after resection.	N/A
Hou et al. (2022) [44]	CRC Serum	Stage II CRC—16 patients with Perineural Invasion and 16 without as well as 16 HC	Size exclusion chromatography, and Total Exosome Isolation Reagent	TMT-LC-MS/MS	SFN	CRC prognosis	1% FDR
Dash et al. (2023) [45]	CRC plasma	Discovery: combination of CRC cell lines and 30 CRC plasma and 30 healthy controls. Validation: 80 CRC (various stages) and 73 HC	Ultracentrifugation	2D-LC-MS/MS for discovery and MRM validation	ADAM10, CD59, TSPAN9	CRC diagnosis	1% FDR, (2 + peptides ≥ 7 aa)
Zhang et al. (2023) [46]	CRC Faeces	Discovery: Bioinformatics screening followed by confirmation on 4 CRC and 4 HC. ELISA validation on 48 CRC patients and 16 HC faecal samples.	Ultracentrifugation	Western blotting	CD147, A33	CRC diagnosis	n/a
Kashara et al. (2023) [47]	CRC Plasma	Discovery phase—26 Stage I (and 26 HC) and 33 Stage IV (and 33 HC);Targeted proteomics—457 proteins. Validation on 141 CRC patients and 9 HC (139 peptides (99 proteins)	Ultracentrifugation	LC-MS/MS and SRM (MS-QBiC)	ORM1	CRC diagnosis and prognosis	1% FDR protein present in all samples at threshold levels by absolute quantitation.
Vallejos et al. (2023) [48]	CRC Plasma	CRC PC (perineural carcinogenesis) (n = 17), CRC VM (visceral metastasis) (n = 17), and CRC NM (non-metastatic) (n = 13)—validation on 9 CRC PC patients followed by Western blotting.	ExoQuick (System biosciences) kit	LC-MS/MS	TLN1, C3	Metastatic-specific exosome signature for prognosis.	1% FDR (protein) and 2% FDR (peptide). 2 unique peptides (7 aa)

AA: advanced adenoma, MS-QBiC: MS-based Quantification By isotope-labelled Cell-free products AHSG: alpha 2-HS glycoprotein, ANXA: Annexin, APOF: Apolipoprotein F, ATIC: Bifunctional purine biosynthesis protein ATIC, BD: benign liver disease, C3: complement C3, CC: colon cancer, CFB: Complement factor B, CRC: colorectal cancer, CRLM: colorectal liver metastasis, CXCL7 Chemokine ligand 7, DIA: data-independent acquisition, FDR: false discovery rate, FGA: fibrinogen α chain, FN1: fibronectin 1, GCLM: Glutamate-Cysteine Ligase Modifier Subunit, HC: healthy controls, HP: haptoglobin, HSP90AA1: heat shock protein 90 alpha family class 1, KEL: Kell blood group glycoprotein, LBP: lipopolysaccharide binding protein, LM: liver metastases, LRG1: leucine-rich alpha-2-glycoprotein 1, ORM1: alpha-1-acid glycoprotein 1, PDE5A: cGMP-specific 3′,5′-cyclic phosphodiesterase, PRM: parallel reaction monitoring, SFN: Stratifin, SPARC: secreted protein acidic and cysteine-rich, TLN1: Talin-1, TMT: tandem mass tagging, TSPAN8: tetraspanin 8, TSPNA1: tetraspanin 1. * MS study was performed on pooled samples; ** validation was performed with Western blotting using independent cohorts.

## 2. Materials and Methods

Ethics statement and plasma sample collection: This study was performed with approval from the Macquarie University Human Research Ethics Committee (MQ HREC approval #5201200702). Clinically staged CRC EDTA plasma (n = 80, 20 patients from each AJCC stage I–IV) and healthy control samples (n = 20) were procured from the Victorian Cancer Biobank (Melbourne, Australia) [28,29] (Table 2). To minimise pre-analytical variation, the plasma samples obtained were stringently age- and sex-matched with strict inclusion/exclusion criteria applied. In detail, the study population was a mixture of near equal groups of females and males (52:48), aged between 50 and 80 years for each CRC stage and for healthy unaffected controls (HCs). Plasmas were collected from CRC patients diagnosed with non-malignant/malignant tumours before they underwent any treatment and/or surgery for CRC. Control unaffected plasma samples were collected from 20 individuals who were aged-matched to the clinical CRC plasma sample cohort and had no apparent evidence of any disease (i.e., no evidence of inflammation or metastatic conditions; no previous history of tumour, cancer, or major therapy). Cancer and healthy plasma samples were de-identified prior to being processed identically throughout this study. All plasma samples were prepared identically, as described previously [29]. Details of clinical features are provided in Appendix A.

### 2.1. Isolation of Plasma EVs and Protein Extraction

Plasma EVs were isolated using a previously published centrifugation method [49] with minor modifications to reduce plasma protein contamination. Plasma samples (500 µL) were centrifuged at 14,000× *g*, 4 °C for 2 min to remove platelets and any solid particles. EVs were then pelleted by centrifugation at 21,400× *g* for 120 min at 4 °C, leaving approximately 10 µL of plasma to avoid disturbing the EV pellet. The EV pellet was washed twice with 1 mL of PBS, each followed by centrifugation at 21,400× *g* for 60 min at 4 °C. These washing steps diluted residual plasma proteins by approximately 10,000-fold.

Proteins were extracted from EV pellets using a whole-cell lysis method [50]. Briefly, EV pellets were lysed with whole-cell lysis buffer (100 mM TEAB, 1% sodium deoxycholate) using a probe sonicator (Branson Sonifier 450; 10 bursts at 40% amplitude, output 2 setting, repeated 3×) followed by heating to 95 °C for 5 min.

### 2.2. Protein Digestion and High-pH Peptide Fractionation

Plasma EV extracted proteins were reduced with 5 mM DTT for 30 min at 60 °C, followed by alkylation with 14 mM iodoacetamide for 30 min at room temperature in the dark. This was followed by digestion using trypsin at a ratio of 1:25 at 37 °C overnight with gentle shaking. Formic acid was then added (1% final concentration) to precipitate sodium deoxycholate and the samples were centrifuged at 12,000× *g* for 5 min (repeated twice). The peptides were then dried on a SpeedVac Vacuum Concentrator (Thermo Fisher Scientific, Massachusetts, USA). Prior to peptides being loading onto the MS, peptides were resuspended with MS loading buffer (0.1% formic acid (*v*/*v*) and 2% acetonitrile (*v*/*v*)), and the concentration of each peptide sample was measured with NanoDrop™ spectrophotometers (Thermo Fisher Scientific, Massachusetts, USA).

For the peptide spectral library generation, digested peptides were fractionated with sequential elution by increasing the ACN concentration (3%, 6%, 9%, 15%, and 80% (*v*/*v*) in 5 mM NH_4_COOH) using the C18 (Empore 2215) stage-tip high-pH fractionation method [51].

### 2.3. Peptide Spectral Library Generation (Data-Dependent Acquisition, DDA)

Fractionated peptides obtained from high-pH peptide fractionation were used to generate a peptide spectral library (i.e., protein identification). The identification was performed on a SCIEX TripleTOF 6600 (SCIEX, Framingham, MA, USA) coupled to an Eksigent Ultra nanoLC system (Eksigent Technologies, Dublin, CA, USA) following a previously described protocol [30]. Briefly, peptides were injected onto a C18 column (packed in-house) for pre-concentration and desalted with MS loading buffer. After desalting, the peptide trap was switched in-line with a cHiPLC C18 column (15 cm × 200 μm, 3 μm, ChromXP C18-CL, 120 Å, 25 °C, SCIEX) and peptides eluted using a linear 120 min gradient from 5% acetonitrile to 35% mobile phase B (B: 99.9% acetonitrile, 0.1% formic acid). In DDA mode, a TOF-MS survey scan was acquired at *m*/*z* 350–1500 with a 0.25 s accumulation time, with the 20 most intense precursor ions (2+–5+; counts > 200) in the survey scan consecutively isolated for subsequent product ion scans. Dynamic exclusion was used with a window of 30 s.

DDA data were analysed using ProteinPilot (V5.0, SCIEX) using the Paragon algorithm [52] against the *Homo sapiens* protein sequence database with reviewed entries obtained from SwissProt. The search parameters were as follows: [cys alkylation: iodoacetamide], [digestion: trypsin], [instrument: TripleTOF 6600], [ID focus: biological modifications], [precursor peptide mass tolerance: ±50 ppm]. A reverse-decoy database search strategy was used with ProteinPilot, with a calculated protein false discovery rate (FDR) < 1% and a probability cut off at 0.99.

### 2.4. SWATH-MS (Data-Independent Acquisition, DIA)

A Sciex TripleTOF 6600 coupled with an Eksigent Ultra nanoLC system and identical LC conditions were used for SWATH-MS experiments [30]. The precursor *m*/*z* frequencies from generated DDA data were used to determine the sizes of the *m*/*z* window. SWATH variable window acquisition with a set of 60 overlapping windows (1 amu for window overlap) was constructed, covering the mass range of m/a 399.5–1249.5. In SWATH mode, TOFMS survey scans were acquired (*m*/*z* 350–1500, 0.05 s), and then the 60 predefined *m*/*z* ranges were sequentially subjected to MS/MS analysis. Product ion spectra were accumulated for 60 milliseconds in the mass range *m*/*z* 350–1500 with rolling collision energy optimised for a lower *m*/*z* in the *m*/*z* window +10%.

### 2.5. SWATH Data Extraction

SWATH data was extracted using PeakView (v2.1) with a SWATH quantitation plug-in (SCIEX). The top 6 fragment ions for each peptide were extracted from the SWATH data using a 75 ppm target XIC width, peptide confidence threshold of ≥0.99, and a 10 min retention time extraction window. After data processing, peptides with confidence > 99% and FDR < 1% (based on chromatographic features after fragment extraction) were used for quantitation. Shared and modified peptides were excluded. The sum of the MS2 ion peak areas of SWATH quantified peptides for individual proteins were exported to calculate the protein peak areas.

### 2.6. Statistical Analyses

Peptide quantification was performed using peak areas from extracted ion chromatograms, and proteins were quantified using cumulative mean values of the calculated peptide quantities. The extracted data was normalised using total area normalisation and log-transformed before statistical analysis; the data distribution was examined using density plots and boxplots. The overall sample look and consistency of the technical replicates was examined visually using hierarchical clustering and principal component analysis (PCA) plots.

Extracted quantitation data emanated from 100 individual samples belonging to five categories (CRC stages I–IV and healthy control) or were from two categories (within only CRC stages I-III, patients who had or had not experienced tumour recurrence inside the 5-year post-surgery period). The GenePattern SWATH workflow [53] was used to determine differentially expressed proteins between groups. This workflow performs ANOVA analyses on log-transformed normalised protein peak areas and peptide-level *t*-tests to determine differential expression. Proteins were considered differentially expressed if they met the criteria of an ANOVA *p*-value less than 0.05 and a fold change greater or less than 1.5 [54,55]. While Benjamani–Hochberg FDR-adjusted ANOVA *p*-values were also generated, they were not used by default. The thresholds applied in this analysis (fold change > 1.5 and *p*-value < 0.05) have been validated through spike-in experiments specific to the SWATH approach and shown to effectively control the FDR [53]. This methodology is further supported by its successful application in other studies [54,55].

## 3. Results

### 3.1. Plasma EV Protein Identification with High-Stringency Protein Inferences

In the past decade, the field of proteomics has transformed and continues to transform many aspects of the molecular sciences [56]. However, discrepancies in data analysis standards have often reduced confidence in some experimental datasets [57]. In order to maintain a high standard of quality and confidence for protein inferences made for the compendium of human proteins observed in our data, we strictly applied the Human Proteome Project (HPP) MS Data Interpretation Guidelines Version 3 [58]. This guideline stipulates that to have the highest confidence in a typical human proteome experiment, each protein group should be identified by a minimum of two peptides, with each peptide having ≥9 amino acids unique to that protein, being non-nested, and being identified with a minimum 1% global FDR. As such, the number of proteins identified was reduced by 37%, from the original 1362 proteins identified using machine default settings down to 853 proteins using these high-stringency HPP requirements (Figure 1). Similarly, 43% of the peptides identified were eliminated from further study upon the application of the criteria (10,933 peptides to 6231 peptides). Appendix A contains detailed information on the identified peptides from each protein at different stringency levels. We note that the majority of recent published proteomic studies use default settings (i.e., 1 ± peptide/protein containing only seven or more amino acids/peptides) to infer protein group identification [57]. These protein inference differences may explain many of the inconsistencies found in biomarker identification from EV studies reported in the previous literature (refer to Table 1).

### 3.2. Evaluation of Plasma EV Isolation and Protein Extraction

To evaluate the plasma EV isolation and protein extraction methodology used in this study, we compared the high-stringency identified EV proteins (from our study) to two significant manually curated databases, Exocarta (17) and Vesiclepedia (18), and also to the more recently released human extracellular vesicle PeptideAtlas 2021-06 database release, in which proteins were identified from 23 experiments derived from five datasets [27]. To maximise plasma EV protein identification and to generate a comprehensive SWATH reference spectral library, we combined healthy and CRC plasma EV protein extracts (n = 100) to cover all proteins present under both healthy and disease conditions. After tryptic digestion of the accumulated EV proteins, we employed high-pH C18 peptide fractionation prior to MS.

A large proportion of the proteins (748 out of 835) in our study have been identified previously in all three EV-specific databases (Figure 2a), with only 16 not having been seen in either database. Of these 16 proteins, 11 emanated from the immunoglobulin family, while the remainder were HLA proteins or isoforms. The largest overlap outside of this was with the high-stringency PeptideAtlas dataset (79 proteins), increasing confidence in the isolation methodology employed in this study. Refer to Appendix A for the lists of proteins in the databases (i.e., Exocarta, Vesiclepedia, and PeptideAtlas).

We further compared our data to the top 100 EV markers obtained through Vesiclepedia and Exocarta, demonstrating that despite the high-stringency protein inference used here, we were able to identify over 76% of the top 100 EV markers (Figure 2b). Additionally, we demonstrate that our methodology was able to pick up several canonical markers for all the different components of the EV (i.e., apoptotic bodies, microvesicles, and exosomes) [24], suggesting that our isolation methodology (i.e., centrifuge at 21,400× *g* for 120 min at 4 °C) (Figure 2c) effectively isolated most EV components.

Finally, we performed a GO analysis (Figure 2d) to determine the cellular components of the isolated proteins found in our study. The majority of proteins originated from the exosome or the cytoplasm, with some expected overlaps with the lysosome, extracellular matrix, and cytoskeleton. This analysis again instilled confidence that the EV isolation methodology we used was both consistent and effective.

### 3.3. Dysregulated Plasma EV Proteins in Early-Stage CRC Compared to Healthy Controls

Following quality control checks on the isolated EV extracts, we conducted quantitative analyses to identify potential diagnostic protein indicators. We focused on early-stage CRC markers (i.e., stage I) due to the high survival rate (up to 95%) associated with early diagnosis [2] and the unmet clinical need for reliable early plasma protein markers. To identify plasma EV proteins differentially expressed between stage I CRC patients and healthy controls, we applied one-way ANOVA and pairwise *t*-tests. All differentially expressed proteins were selected based on a *p*-value ≤ 0.05 and a fold change ratio cut-off of ±1.5. This analysis resulted in the identification of a total of 11 proteins that exhibited differential (↓↑) expression in CRC stage I patients when compared to healthy controls (Figure 3a). Expression measurements for all quantified proteins (normalised) at each disease stage (including controls) are provided in Appendix A. Volcano plots representing differentially expressed proteins in other stages (II, III, and IV) compared to healthy control are provided in Appendix A.

Of the proteins that met the selection criteria outlined above, three proteins were significantly up-regulated (ALDH9A1, F9, and SMIM1), whilst eight proteins (PFKP, MDH2, CYRIB, LRRFIP1, HPR, ACTN2, C1QC, and ADIPOQ) were shown to be significantly down-regulated in stage I compared to healthy controls (Figure 3a). A search of these dysregulated proteins in EV protein databases (ExoCarta, Vesiclepedia, and PeptideAtlas) confirmed that all are compiled in these databases, indicating their prior detection in EV studies. Of the significantly up-regulated proteins, ALDH9A1 demonstrated a 2-fold increase in expression in stage I; however, it was not statistically significant in stage II despite remaining elevated, suggesting a potential return to normal levels. Similarly, C1QC was significantly down-regulated (FC 2.62) in stage I but did not demonstrate significance in stage II, though its levels appeared to return to normal levels. These observations may hold translational value for understanding CRC progression, especially in stage II CRC where the risk of tumour recurrence remains a clinically unmet need [3]. Additionally, ADIPOQ, ACTN2, and MDH2 were consistently down-regulated across both stages. Detailed information on fold changes and *p*-values for all differentially expressed proteins can be found in Appendix A.

### 3.4. Dysregulated Plasma EV Proteins Between CRC Patients Who Had or Had No Tumour Recurrence—Predict the Risk of Tumour Recurrence

Since 5–33% of CRC patients diagnosed with stages I, II, or III experience recurrence/relapse within 5 years of primary tumour resection [2,3], identifying prognostic biomarkers to predict tumour recurrence risk is crucial. At earlier stages (I/II), adjuvant treatment is often not administered due to the lack of histological evidence of metastasis to lymph nodes or distant sites. Consequently, if CRC recurs, it is frequently detected at a more advanced stage, where survival rates are significantly lower.

In our analysis, stage IV patients were excluded due to their existing distal metastasis. Within the remaining cohort, 2 out of 20 (10%) stage I, 4 out of 20 (20%) stage II, and 7 out of 20 (35%) stage III patients (Table 2) experienced tumour recurrence within 5 years post-surgery. Overall, combining stages I–III, 13 patients exhibited tumour recurrence (recur group), while 47 patients showed no recurrence (non-recur group) within the 5-year post-surgery period.

We applied identical protein inference stringency analysis as in the initial study and demonstrated that 11 proteins (i.e., GDI1, SLC2A1, DARS1, ENO2, PGM1, COL6A1, LRRFIP2, NSF, TMED9, LTB4R, and CO9) were significantly up-regulated in patients whose tumour recurred compared to those whose tumour did not. Similarly, three proteins were observed to be significantly down-regulated (i.e., TSG101, IQGAP1, and CAMK1) in relapsing patients, with CAMK1 demonstrating a greater than 4-fold change (Figure 4a,b). Expression levels for all quantified proteins between the recurred and non-recurred groups are provided in Appendix A.

## 4. Discussion

Plasma EV proteins offer an exciting frontier for discovering disease markers, yet their application in diagnostics and prognostics brings substantial challenges. These challenges primarily relate to experimental methodology and the clinical relevance and applicability of the findings, especially in rigorous clinical environments [21]. Here, we discuss key aspects of our experimental approach, covering both methodological considerations and practical challenges. We believe this work expands our understanding of the proteomic landscape of plasma EVs, marking a crucial step toward identifying early-stage CRC biomarkers and exploring potential links between the plasma EV proteome and tumour recurrence.

### 4.1. EV Isolation from Plasma

The two most prominent databases (Exocarta and Vesiclepedia) represent a comprehensive repository of EV proteins. These come from hundreds of experiments undertaken in different labs using different experimental methodologies and samples (including cell lines and tissue samples) and reports on mRNA and lipids from extracellular vesicles. Most, if not all, proteomics studies (Table 1) compare their datasets to these two central databases (as we have). The value in comparison is to demonstrate the validity of experimental methodologies. Indeed, we demonstrated a significant overlap with previously reported proteins (Figure 2). However, 16 were not seen in any of the other databases. Equally, 71 (Exocarta), 6134 (Vesiclepedia), and 210 (PeptideAtlas) proteins were not observed in our dataset. Admittedly, the larger dataset from 100s of experiments reported in these databases would not possibly cover the size of our dataset. Many of those protein groups (37%) were eliminated from our analysis due to data being of low stringency. The proteins reported in Exocarta and Vesiclepedia contain proteins derived mostly from default stringency data (often ≥7 aa, no unitypicity information, and >1% FDR), so we contend that these differences are understandable. However, what was helpful was that we were able to demonstrate peptides (proteins) of all major classes of EVs (i.e., apoptotic bodies, microvesicles, and exosomes) (Figure 2c) in our stringent dataset in all 100 samples examined. This finding allowed a high level of confidence that most of the proteins we extracted were EV-related.

We also note that despite the various methodologies used by other researchers (Table 1), centrifugation of the plasma at 21,400× *g* for 120 min at 4 °C was sufficient to obtain a relatively pure sample of EVs. Critically, this finding suggests that such a methodology may be more suitable and amenable to clinical application whilst still providing the resolution necessary to identify markers of disease. Additionally, size exclusion chromatography (SEC) and sucrose gradient ultracentrifugation have been used to isolate EVs and are touted to be the methods of choice. Both certainly have their advantages in terms of yield and quality [59,60].

### 4.2. EV Proteins Associated with Early CRC Stage

The need to uncover robust molecular tumour stage markers is a known and accepted unmet need [61] clinically, with multiple studies (Table 1) working toward such a goal. In many cases of the disease, the limitation of survival and prognosis for a patient is attributable to delayed diagnosis [62]. In this study, we employed a relatively small cohort of 80 patients equally from all four stages of CRC, using a robust quantitative proteomics approach (SWATH-MS) to uncover dysregulation in EV protein expression. We focused on a biomarker’s ability to discriminate early stages I and II CRC from healthy controls.

The dysregulated proteins seem to fall into three broad categories. The first involves proteins in energy metabolism and regulation, PFKP (ATP-dependent 6-phosphofructokinase), MDH2 (Malate dehydrogenase), and ADIPOQ (Adiponectin receptor protein), which were all down-regulated, and ALDH9A1 (4-trimethylaminobutyraldehyde dehydrogenase), which was significantly up-regulated. The dysregulation of the glucose metabolism pathways in cancer is well known [63], suggesting that these early changes in metabolism are captured in the EV proteome. The down-regulation of PFK and MDH2 could be an early indicator of cellular stress. However, it is not clear what the up-regulation of ALDH9A1 means.

The second category concerns proteins involved in cellular remodelling and the cytoskeleton or its regulation, CYRIB (CYFIP-related Rac1 interactor B), LRRFIP1 (Leucine-rich repeat flightless-interacting protein 1), and ACTN2 (Alpha-actinin-2), all of which were down-regulated. Again, being a hallmark of cancer [64], remodelling of the cytoskeletal architecture and its regulation may be captured in the EV proteome. However, a protein decrease in cellular modelling seems contrary to the prevailing understanding that cellular remodelling is increased as cancer progresses [65]. It may be that the observed changes are systemic responses to the presence of the tumour rather than tumour-specific ones.

Finally, the third categorisation is that of dysregulated proteins that generally reside in the blood or have immune-related functions. These include HPR (Haptoglobin-related protein) and C1QC (Complement C1q subcomponent subunit C), both of which were significantly down-regulated, and F9 (Coagulation factor IX) and SMIM1 (Small integral membrane protein 1), which were up-regulated. Interestingly, the significant up-regulation of SMIM1, a regulator of red blood cell formation [66], can be correlated to anaemia, which normally accompanies early CRC diagnosis [67]. Whether these changes are system-wide or localised to the tumour remains to be elucidated.

### 4.3. EV Proteins Associated with CRC Tumour Recurrence

The fundamental question of tumour recurrence post-surgical intervention is also a clinically significant unmet need [68], prompting many studies to investigate EVs, although the majority focus on DNA-based markers [69]. Our study is one of the few that specifically investigated CRC recurrence markers using proteomics. This study specifically investigated the markers of recurrence in stages I, II, and III for two primary reasons. Firstly, no sample was available to assess relapse in stage IV; secondly, even if there was, the survival rate in patients diagnosed with stage IV is too low to justify risk stratification.

The proteins confidently found to be dysregulated in the patients whose tumours recurred vs. those whose tumours did not recur fell into four broad categories. Firstly, proteins related to the regulation or formation of EVs, such as the up-regulation of GDI1- Ras-related protein Rab-5A, a protein involved in vesicular trafficking and a suspected marker of oral cancer [70], NSF (Vesicle-fusing ATPase), and TMED9 (Transmembrane emp24 domain-containing protein 9), a protein shown to oppose signalling that promotes metastasis in CRC [71]. Equally, the down-regulation of TSG101 (Tumour susceptibility gene 101 protein), which regulates vesicular traffic [72], seems to be an often-cited protein in cancer-related EV studies [73]. The implications of such a discovery in patients who are more susceptible to relapse are intriguing. They may lend credence to the notion that EVs may be involved in the biology of ‘micro-metastasis’ [61], whereby those more susceptible to relapse may be those that have systemic abnormalities in EV regulation that are manifested more prominently when exposed to a tumour.

The second category of dysregulated proteins includes those involved in cytoskeleton remodelling and associated signalling pathways. Of the significantly up-regulated proteins was COL6A1 (Collagen alpha-1 (VI) chain), LRRFIP2 (Leucine-rich repeat flightless-interacting protein 2), an activator of the canonical Wnt signalling pathway, and ENO2 (Gamma-enolase), an acknowledged receptor for the zymogen plasma protease plasminogen. However, IQGAP1 (Ras GTPase-activating-like protein IQGAP1), which is a protein responsible for regulating the dynamics of cytoskeleton remodelling and has been implicated in multiple cancers [74], and CAMK1 (Calcium/calmodulin-dependent protein kinase type 1) were both shown to be down-regulated. As discussed earlier, this phenomenon of cytoskeleton remodelling is as expected from the literature, and it may be that the proteins observed are likely shed from a tumour or immune-derived cells, though there is no real way to assess that in this study.

The third category involves proteins in metabolic processes or the transport of products of metabolism. Proteins such as SLC2A1 (Solute carrier family 2), facilitated glucose transporter member 1 (Glucose transporter basal glucose uptake), PGM1 (Phosphoglucomutase-1), and DARS1 (Aspartate—tRNA ligase) were all up-regulated.

Finally, the last category is those related to the immune response or its regulation. Both LTB4R (Leukotriene B4 receptor 1), a G-protein receptor for leukotriene B4, a potent chemoattractant involved in inflammation and immune response, and CO9 (Complement component C9) were shown to be significantly up-regulated. Although plausible explanations exist for the dysregulation of these latter two categories, the implications of these findings must be further investigated, as any comment would be speculatory. Interestingly, nearly all dysregulated proteins identified are integral membrane proteins; however, their specific origins (tumour-related or otherwise) remain to be investigated.

### 4.4. Discrepancies Across Previous Studies

It is clear that few, if any, of the markers we uncovered in this study correlate with those discovered in other EV studies using CRC plasma or serum (Table 1). In some cases, this discrepancy could be explained by differences in stringency cut-offs of protein inference. In most cases, those studies did not use minimum peptide inference guidelines elucidated by the HPP [58]. Less stringent cut-offs may lead to higher false positive results. The interpretation guidelines recommend that two nine-amino acid, unitypical, non-nestled peptides be used to infer the identity of a protein, amongst other guidelines. Although our study ensured robustness by eliminating less reliable results, most other studies outlined in Table 1 did not do so in the discovery phase of experiments. However, subsequent validation (for the studies that did validate) of the markers did suggest robust findings.

Additionally, the selection of a particular EV isolation method can significantly impact the outcomes of biomarker discovery studies. These methods exhibit variations in selectivity towards specific EV biologies, protein classes, or protein isoforms. For instance, a range of EV protein isolation techniques were employed, as indicated in Table 1. These techniques encompass sucrose gradient centrifugation, ultracentrifugation, SEC, and several commercially available EV isolation kits. Whilst each method has its advantages and limitations, the consistency of protein isolation can differ, leading to varying results. The choice of isolation method directly influences the composition of the isolated EV population, potentially biassing subsequently analysed proteins [75]. Therefore, utilising different EV isolation methods across studies can contribute to discrepancies in identified protein biomarkers. For instance, a study by Dong et al. demonstrated significant differences between the three most common separation methodologies: ultracentrifugation, precipitation, and size exclusion chromatography [76]. More recently, a review of immune capture methodologies also discussed the potential for significant proteomic differences based on isolation methodology [77]. However, the impact of many of the methodologies, such as differential centrifugation [78], density gradient fractionation, or a combination [79], is yet to be carefully evaluated, as each present confounding factors that may significantly alter the proportion of subpopulations of EVs isolated. We contend that for a catch-all downstream methodology such as SWATH-MS, a crude ’general’ EV isolation methodology such as centrifugation is likely to be more effective in ensuring all relevant subpopulations of EVs are captured. Interestingly, none of the previous studies on CRC used our methodology, which confounds a direct comparison of the results.

Finally, many studies involve small patient cohorts, which may contribute to a lack of correlation in findings. However, even larger studies [43,45,47] with sample sizes comparable to ours have shown limited alignment in the identified markers. This represents a microcosm of the challenges associated with discovering biomarkers from EVs. Of these, one of the most obvious is the variability of EVs between samples and patients at any given moment. The nature, source, target, and regulation of the various EV types are yet to be elucidated in detail [80], though it is clear that a significant amount of variability exists both between subjects and within subjects [81], fuelled primarily by the half-life of EVs [82] and other immune functions. Additionally, it would be unsurprising if other variables such as nutrition, culture, weather geography, and other factors also play a role in this inherent variability. It is well acknowledged that the lack of standardised protocols (with hundreds of pre-analytical protocols and over 40 variables) in EV isolation [83] remains a significant challenge. Carefully controlled studies on large cohorts at various time points, with appropriate controls [84,85] would be required to overcome such limitations to uncover robust disease markers. Nevertheless, each study in its own right builds a unique picture of the biology of CRC, especially regarding system-wide effects.

## 5. Conclusions

This study uncovered a robust and reproducible experimental protocol that enables comprehensive profiling of extracellular vesicle (EV) proteins with minimal loss, making it suitable for routine laboratory implementation. Our findings shed light on the fundamental biology of CRC, particularly the roles of EV proteins in cellular remodelling and metabolic dysregulation during disease progression and recurrence. Importantly, the results suggest that micro-metastasis may occur earlier than previously anticipated, potentially driven by EV-mediated processes, which warrants further investigation.

While this study highlights the potential of EV proteins as biomarkers for CRC, several limitations must be addressed. The inherent variability in EV profiles among individuals underscores the need for larger and more diverse cohorts, incorporating variables such as age, gender, geography, and the timing of sample collection. Additionally, standardising EV isolation and proteomic analysis protocols is essential for improving reproducibility across studies. Future research should focus on validating these findings in independent cohorts and integrating orthogonal validation techniques to enhance biomarker reliability. These efforts will be critical for advancing the clinical application of EV-based biomarkers in early CRC detection and recurrence monitoring.

## Figures and Tables

**Figure 1 cancers-16-04259-f001:**
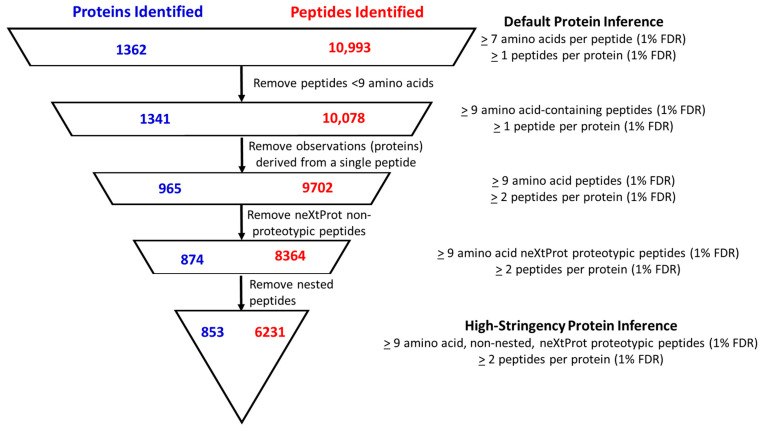
The systematic application of high-stringency criteria to the identification of proteins for this study, resulting in a dataset of 853 proteins of high confidence. We observed a reduction of 37% in the number of proteins and 43% in that of peptides compared to the default settings. Proteolytic peptides are defined as those that are consistently identified by MS and uniquely identify each protein. A nested peptide is an identified peptide sequence that is fully subsumed within another identified peptide sequence. Appendix A details the peptides identified for each protein across different stringency levels.

**Figure 2 cancers-16-04259-f002:**
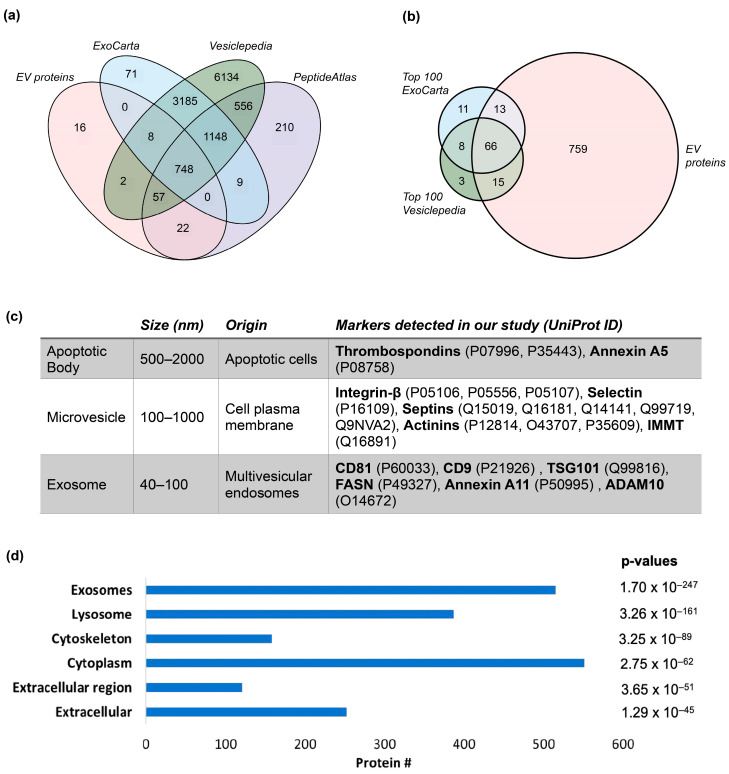
CRC/healthy plasma extracellular vesicle (EV) protein identification. Venn diagram comparisons between (**a**) EV proteins identified from our study and extracellular vesicle protein databases from ExoCarta, Vesiclepedia, and Human EV PeptideAtlas [27] and (**b**) EV proteins and top 100 extracellular vesicle protein markers from ExoCarta and Vesiclepedia. (**c**) Protein markers identified from our study represent components of EVs, including the apoptotic body, microvesicle, and exosome [24]. (**d**) Cellular component Gene Ontology (GO) analysis of identified EV proteins. ExoCarta and Vesiclepedia EV protein databases downloaded from http://www.exocarta.org/ (accessed on 17 December 2024) and http://microvesicles.org/ (accessed on 17 December 2024), respectively. Appendix A provides detailed information, including the lists of proteins in each database, the detection methodologies employed, and the data accessed dates.

**Figure 3 cancers-16-04259-f003:**
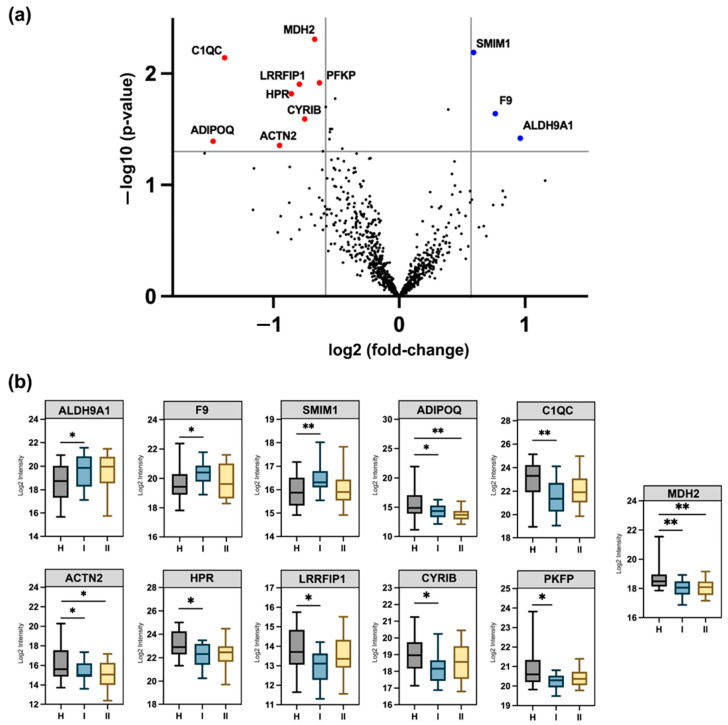
Plasma EV protein quantification in early stage I of CRC vs. healthy controls. (**a**) Volcano plot representations on differentially expressed proteins (FC > 1.5, *p*-value < 0.05) between stage I and healthy controls. Blue dots indicate up-regulated proteins and red dots indicate down-regulated proteins in stage I compared to controls. (**b**) Box plots illustrate the protein expression patterns between control, stage I, and stage II. *: *p*-value < 0.05, **: *p*-value < 0.01.

**Figure 4 cancers-16-04259-f004:**
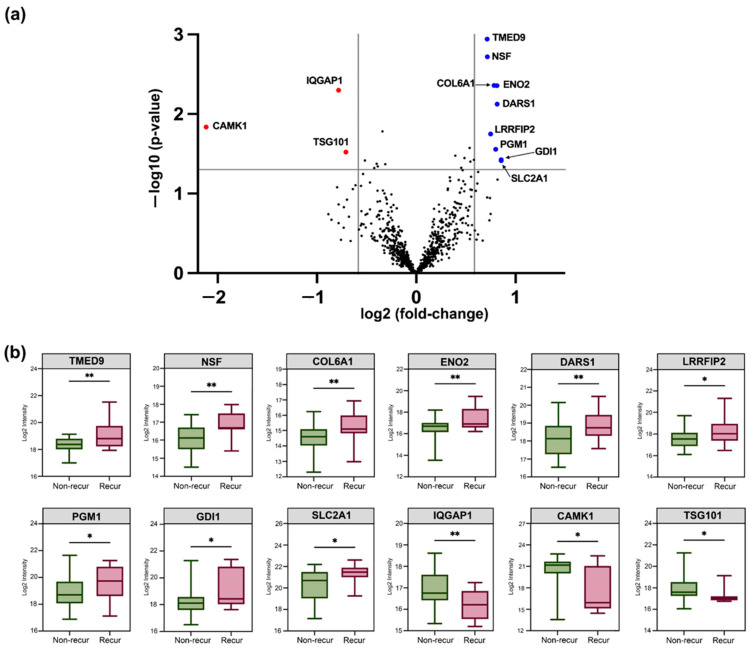
Plasma EV protein quantification comparing the non-recur group (47 CRC patients in stages I/II/III without tumour recurrence within 5 years of primary tumour resection) and recur group (13 CRC patients in stages I/II/III with tumour recurrence within 5 years). (**a**) Volcano plot representations of differentially expressed proteins (FC > 1.5, *p*-value < 0.05) between non-recurred and recurred patient groups. Blue dots indicate up-regulated proteins and red dots indicate down-regulated proteins in recurred compared to non-recurred (i.e., cured). (**b**) Box plots illustrate the protein expression patterns between the non-recur and recur groups. *: *p*-value < 0.05, **: *p*-value < 0.01.

**Table 2 cancers-16-04259-t002:** Clinical features (five-year overall survival and recurrence rates) of recruited CRC patient cohort (HC: healthy control; OS: overall survival).

Variable	Category	HCs (n = 20)	Stage I (n = 20)	Stage II (n = 20)	Stage III (n = 20)	Stage IV (n = 20)
Age	Median ± SD	63.5 ± 7.8	63.5 ± 8.1	70.5 ± 7.5	60.5 ± 8.3	63.0 ± 7.6
Gender	Male (%)	10 (50)	10 (50)	10 (50)	9 (45)	9 (45)
	Female (%)	10 (50)	10 (50)	10 (50)	11 (55)	11 (55)
5-yr recurrence Postoperative	Yes (%)	-	2 (10)	4 (20)	7 (35)	-
	No (%)	-	18 (90)	16 (80)	13 (65)	-
5-yr OS Postoperative	Yes (%)	-	20 (100)	16 (80)	13 (65)	6 (30)
	No (%)	-	0 (0)	4 (20)	7 (35)	14 (70)
Postoperative chemotherapy	Yes (%)	-	1 (5)	1 (5)	17 (85)	8 (40)
	No (%)	-	19 (95)	19 (95)	3 (15)	12 (60)

## Data Availability

All data are available in the Appendix A.

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
