# Peer review of "A Proteomic Examination of Plasma Extracellular Vesicles Across Colorectal Cancer Stages Uncovers Biological Insights That Potentially Improve Prognosis"

_cancers, 2024, doi:10.3390/cancers16244259_

Round 1

Reviewer 1 Report

Comments and Suggestions for Authors

Abstract:

1. The abstract highlights advancements in the understanding of plasma extracellular vesicles (EVs) without specifying these progressions. Including a brief mention of key discoveries or concepts would enhance clarity.

2. Ensure all abbreviations (e.g., CRC for colorectal cancer) are defined upon first usage.

3. The introduction must articulately present the hypothesis or research inquiries guiding the study. This is crucial for establishing the foundation for subsequent methods and outcomes.

Methods:

4. The methodology should explicitly state the rationale for the sample size and the statistical techniques employed for analysis. If referencing high-resolution SWATH-MS, provide an explanation of its advantages over alternative methods.

5. Verify the accuracy and consistency of all technical terms (e.g., "centrifuge" should undergo spell-check).

6. Specify the statistical tests used for data analysis in the methods section, including any adjustments for multiple comparisons. This is fundamental for interpreting the significance of the results.

Results:

7. The results section should present precise data on discoveries, encompassing specific proteins identified and their connection to CRC stages. Clearly state any reported statistical significance.

8. Review the text for any missing figures or tables. Ensure all shapes are correctly labeled and align with the text.

Discussion:

9. The discussion should critically assess the findings alongside prior studies on EVs in CRC. Address discrepancies, if any, and explore potential reasons for such distinctions.

10. Investigate the biological mechanisms underlying the discerned alterations in protein expression in the discussion. Explain how the identified proteins may contribute to tumor progression or recurrence.

11. Propose future research directions based on the findings in the discussion. This could encompass plans to validate the identified biomarkers in broader cohorts or probe into their functional roles in CRC.

12. Analyze the findings critically within the context of existing literature in the discussion. If the study suggests earlier occurrence of micrometastasis than previously understood, substantiate this claim with evidence or reputable sources.

Conclusion:

13. Summarize the main findings and their implications for future research or clinical application in the conclusion. If limitations are discussed, offer specific and constructive insights.

14. Ensure clarity and conciseness in the conclusion, avoiding overly intricate sentences that may obscure comprehension for readers.

Comments on the Quality of English Language

The English could be improved to more clearly express the research.

Reviewer 2 Report

Comments and Suggestions for Authors

Review on "A proteomic examination of plasma extracellular vesicles across colorectal cancer stages uncovers biological insights that potentially improve prognosis" by Mohamedali et al.

This report carried out a proteomics study analyzing extra vesicles obtained from colorectal cancer (CRC) patients. Despite the small number of samples per pathophysiological group, the study was able to detect differentially regulated proteins in the early stage through the late stage of CRC relative to normal specimens. The success may be attributed to the meticulous grouping that considered the age and gender as factors. The interpretation of the data is well-balanced with discussion of the methodology, the guidelines for EV analyses, and comparisons with the previous studies. The data should help facilitate further studies toward establishing CRC early biomarkers based on EV. Minor comments are raised as below.

---------

The analysis was carried out with a small number of sample sets (total 100, 20 per group of normal and Stage I, II, III, and, IV). The 20 per group was further split to male/female, and so the analyses were done with just 10 specimens per group. Did this study employ a prior estimation of the sample number? If so, how was this estimate carried out (what was the null-hypothesis; what significant level and power values were set)?

Line 187-189: the description is not consistent with the Fig. 3. ALDH9A1 level returned to normal levels in stage II CRC, but in Fig. 3, its level stays as high as that of stage I. Similarly, C1QC returned to the normal level in stage II, but ADIPOQ showed further decrease in stage II from stage I. The observation that some of the proteins returned to the normal level may have significant translational values, but based on Fig. 3b, there does not seem to have statistical significance. This should be pointed out in the body text.

Fig. 3a: The p-values are fairly high, i.e., reaching to 0.05 border line and there are 11 data were shown. P-values and their adjusted p-values (e.g., FDR) should be shown in a table.

Fig. 4: Sample numbers for 'Non-recur' and 'Recur' (47 vs 13) should be written in the figure 4 legend. Do these proteins in Fig. 4b show the same trend of up/down-regulation in EV data obtained from stage IV specimens? Were p-values evaluated by Welch's t-test? 

Could some of the proteins, e.g., C1QC, be detected due to contamination from the plasma? The authors should discuss this possibility and how it was excluded in addition to the fact that these are regularly found in other studies/databases.

Round 2

Reviewer 1 Report

Comments and Suggestions for Authors

Dear Editor,

I am writing concerning the comments on the present manuscript. All comments have been successfully addressed and resolved, and I believe the manuscript is now ready for acceptance.

Thank you for your time and attention to this matter.

Best regards,